# The Advantages of Non-Adhesive Gel-like Embolic Materials in the Endovascular Treatment of Benign Hypervascularized Lesions of the Head and Neck

**DOI:** 10.3390/gels9120954

**Published:** 2023-12-04

**Authors:** Andrey Petrov, Arkady Ivanov, Egor Kolomin, Nikita Tukanov, Anna Petrova, Larisa Rozhchenko, Julia Suvorova

**Affiliations:** 1Vascular Neurosurgery Department, Polenov Neurosurgical Research Institute, Branch of Almazov National Medical Research Centre, 191014 Saint Petersburg, Russia; arkady.neuro@gmail.com (A.I.); egor96kolomin@gmail.com (E.K.); nikita.tukanov@gmail.com (N.T.); petrovaanna2803@gmail.com (A.P.); rozhch@mail.ru (L.R.); juliavsuvorova@mail.ru (J.S.); 2North-Western District Scientific and Clinical Center Named after L. G. Sokolov Federal Medical and Biological Agency, 194291 Saint Petersburg, Russia

**Keywords:** embolic agent, ONYX, SQUID, PHIL, vascular malformations, paragangliomas

## Abstract

Objectives: The use of non-adhesive gel-like embolic materials (NAGLEMs) in the endovascular treatment of hypervascularized formations in the head and neck is gaining in popularity because of a number of important characteristics involved. Their primary benefits are their capacity to penetrate diseased vasculature, effectively distribute, and, most importantly, remain controllable during the process. We reviewed the literature and evaluated the results of using NAGLEMs in comparison to other embolizing substances (namely, coils, glue, and particles) as alternative embolizing agents for patients receiving care at our clinic. The process comprised evaluating the safety, effectiveness, and technological elements of endovascular therapy used to treat two categories of hypervascular pathological abnormalities that were surgically corrected between 2015 and 2023. Arteriovenous malformations (AVMs) located in the head, neck, and paragangliomas with jugular/carotid body localization are combined by intense shunting blood flow and shared requirements for the embolic agent used in endovascular treatment (such as penetration, distribution, delayed polymerization, and controllability). An analysis of the literature was also conducted. Results showed 18 patients diagnosed with neck paragangliomas of the carotid body and jugular type. Five patients with arteriovenous malformation (AVM) of the face and neck were included, consisting of sixteen females and seven males with an average age of 55 ± 13 years. Endovascular procedures were performed using NAGLEMs (ONYX (Medtronic, Irvine, CA, USA), SQUID (Balt, Montmorency, France), and PHIL (Microvention, Tustin, CA, USA)) and dimethyl sulfoxide (DMSO)-compatible balloon catheters. All patients achieved complete or partial embolization of hypervascularized formations using one or more stages of endovascular treatment. Additionally, three AVMs of the face and two paragangliomas of the neck were surgically excised following embolization. In other instances, formations were not deemed necessary to be removed. The patients’ condition upon discharge was assessed by the modified Rankin Scale (mRs) and rated between 0 and 2. Conclusion: Currently, NAGLEMs are predominantly used to treat hypervascularized formations in the neck and head due to their fundamental properties. These properties include a lack of adhesion and a delay in predictable polymerization (after 30–40 min). NAGLEMs also exhibit excellent distribution and penetration throughout the vascular bed of the formation. Adequate controllability of the process is largely achieved through the presence of embolism forms of different viscosity, as well as excellent X-ray visualization.

## 1. Introduction

Minimally invasive procedures are becoming increasingly popular in the management of benign hypervascularized masses located in the head and neck regions [1,2,3,4,5]. Embolization is a method of blocking the blood supply to these masses, and can be implemented for preparation before open surgical removal [5,6,7], to decrease blood supply prior to radiosurgery [8], or as a standalone treatment option [9].

Various embolic agents have been proposed to occlude the vascular bed of hypervascularized benign head and neck masses (PVA [6], microspheres [6], microcoils [6], gelatin sponge [6], absolute ethanol [9], ONYX [4]). These embolic agents differ not only in their physical and chemical properties, but also in their intended applications and techniques of delivery into the vascular bed.

Traditionally, embolic agents have been classified into two primary categories based on their aggregate state: liquid and solid. Gels and gel-like substances are mentioned in few studies. The ability to adhere is the primary chemical property used for the classification of embolic materials [10]. Consequently, these materials can distribute evenly throughout the vascular bed, which leads to better filling of pathological vessels. Furthermore, this feature allows for longer and more controlled administration [11]. It should also be noted that the terms “gel” and “gel-like” are applied to non-adhesive embolic agents at different stages of their polymerization and correspond to their physical and chemical properties. The formation of embolic casts is believed to occur through precipitation, which can be influenced by environmental conditions such as gelation. The prevailing mechanism responsible for the dissemination of embolic material in biological tissues remains uncertain. Non-adhesive gel-like embolic materials (NAGLEMs) are a group of biocompatible copolymer compositions that dissolve in dimethyl sulfoxide (DMSO). When in contact with blood, they solidify into a sponge-like gel, forming a rigid, plastic-like cast [11,12,13]. Considering the lack of a clear understanding of the sol–gel transition conditions in biological tissues, these embolic materials can rightly be classified as gel-like, apparently forming at various stages structures similar to thermoreversible xerogel.

The field of vascular neurosurgery has been transformed by the advent of non-adhesive gel-like embolic materials (NAGLEMs) [14]. New treatments for patients with cerebral pathology [15,16,17] and hypervascular formations in the head and neck, specifically paragangliomas and arteriovenous malformations (AVMs), have arisen. The emergence of these non-adhesive compositions is due to the elimination of the shortcomings associated with earlier generations of embolic agents [14]. NAGLEMs, due to their fundamental properties, are becoming increasingly prevalent in the endovascular treatment of hypervascular formations in the head and neck region. Their primary advantages include lack of adhesion, effective distribution and penetration through the pathological formation vessels, and, crucially, adequate process controllability [14].

Some of the most widely recognized and esteemed NAGLEMs comprise ONYX (Medtronic), SQUID (Balt), and PHIL (Microvention) [5,11,12,13,15,16,17,18].

ONYX is undoubtedly the longest used NAGLEM, having first appeared in medical publications in the 1990s [13,19,20,21].

SQUID was initially created for utilization in cerebral vessels. However, it has been used in a growing number of applications in recent times [11,13,15,16,17,22].

ONYX and SQUID are provided in vials. Each vial contains 1.5 mL of embolic agent, which should be shaken for 20 min before use according to the manufacturer’s instructions [11,13,23].

I. Kuianova et al. [11] carried out rheological measurements on two NAGLEMs (ONYX at 18 cps and SQUID at 12 cps). Their analysis indicates that viscosity is affected by both the shear modulus and temperature. Furthermore, the discovered non-monotonicity in the correlation between viscosity and shear rate provides new insight into the non-random nature of this phenomenon. In particular, the algorithm for activating viscosity, which includes adjusting the viscosity of the embolic polymer, grants a qualitative perception of the impacts taking place during embolization.

PHIL appeared later than the previous NAGLEMs and there is limited literature on it [13,23,24].

The characteristics that define the behavior of endovascular embolic agents include their ability to penetrate tissue [13], distribution within the vascular bed, controllability, and better visualization. The term “penetration capacity” refers to the depth of distribution of embolic agents within the vascular bed, ranging from large to small diameter vessels and extending up to the capillary bed, with arterial to venous bed penetration [13]. “Distribution” refers to the even spreading of the embolic material. These characteristics vary according to the type of embolic agent and are based on its chemical–physical properties. Ultimately, these characteristics guarantee the controllability of embolization.

In this article, we analyzed the characteristics of different embolic agents that were used prior to the introduction of NAGLEMs. Subsequently, we make a direct comparison between NAGLEMs and these agents, taking into account evidence from the worldwide literature and our own investigations. We aim to address why NAGLEMs are the preferred choice for treating hypervascularized diseases of the head and neck using intravascular treatment. This location for the pathological process was not chosen arbitrarily. This is because, in addition to neurosurgeons, specialists in maxillofacial [25], plastic [1], vascular [2], and radiosurgery [8] also deal with this condition. This site, where there can be hypervascularized pathological masses, often poses a challenge in terms of treatment tactics.

## 2. Results

### 2.1. Patient Population

In our clinics, specifically the North-Western district scientific and clinical center named after L. G. Sokolov Federal Medical and Biological Agency & Almazov National Research Medical Center in St. Petersburg, Russia, we performed operations on 23 patients with hypervascularized masses in the region of the head and neck between November 2015 and May 2023. Eighteen patients were diagnosed with paragangliomas: seven (30.4%) had carotid body type while eleven (47.8%) had jugular type. Five (21.7%) patients had arteriovenous malformations in their face and neck. Among the patients, sixteen (69.6%) were females and seven (30.4%) were males. In six patients (26.1%), our clinic attempted embolization with non-gel-like embolic agents prior to treatment. However, only partial embolization was observed in all cases. Four patients (17.4%) received microsurgical removal of the mass after embolization. The demographic data and treatment characteristics of patients in both groups are available in Table 1. (Table 1)

NAGLEMs (ONYX (Medtronic), SQUID (Balt), and PHIL (Microvention)) were used. Embolization of the masses was achieved using balloons that are compatible with dimethyl sulfoxide (DMSO) to prevent the embolic agents from migrating into the main vessels. Therefore, the technique was effective in avoiding the risk of migration.

ONYX and SQUID compositions contain ethylene vinyl alcohol copolymer (EVOH), dimethylsulfoxide (DMSO), and tantalum. The main difference between these two NAGLEMs is the size of the tantalum particles and the presence of varying viscosities.

EVOH consists of 48 mol/L ethylene and 52 mol/L vinyl alcohol. ONYX 6.0%, 6.5%, and 8.0% have viscosities of 18, 20, and 34 centipoise (cps; unit of viscosity) or mPa·s, respectively, while the SQUID range has a viscosity of 12 cps (or mPa·s).

PHIL consists of two copolymers, polylactide-co-glycolide and polyhydroxyethylmethacrylate, as active components, DMSO, and triiodphenol (iodine compounds), the latter covalently bound to two copolymers, which ensures the radiopacity of the agent. PHIL’s ready-to-use formula is a potential advantage, avoiding time-consuming preparation. PHIL, which does not contain tantalum, results in fewer CT artefacts than ONYX and SQUID. Moreover, there is no tattooing effect detected if the pathological mass is in proximity to the skin.

SQUID was used in 11 out of 23 cases, ONYX in 8 out of 23, and PHIL in only 1 patient.

The average embolization time was 97 ± 28 (M ± SD) minutes (min 60, max 170). Complete embolization was defined as filling the vascular network of the mass by 100%, while subtotal embolization was defined as 85–99% filling of the vascular network. All 23 cases demonstrated either total or subtotal filling of the vascular network, with 17 neoplasms (73.9%) and 6 subtotals (26.1%) completely turned off. The 100% complete AVMs were turned off in all 5 cases, and para-gangliomas were turned off in 12 out of 18 cases (66%). All patients underwent MRI in the early postoperative period.

The mRS scores at discharge were 0 in 11 patients, 1 in 10 patients, and 2 in 2 patients. As follows from Table 1, the group of patients with AVMs of the face and neck (*n* = 5) was significantly younger (*p* = 0.033) (method used: Fisher’s F-criterion) and the mean age of patients in this group was 42 ± 6 (M ± SD). In this group, embolization attempts with other non-gel-like agents (cyanoacrylates, alcohol, and spirals) were significantly more frequent (*p* = 0.004; method used: Pearson’s chi-square) (Figure 1). 

Despite this, this group significantly required less time (*p* = 0.035; method used: Kruskal–Wallis criterion) to administer NAGLEMs, averaging 70 min (Me) (Figure 2). 

In all cases in this group, total embolization of AVM was performed using DMSO-compatible balloons. The average mRS in this group, both at the time of admission and at the time of discharge, was 0 (Me).

The group of paragangliomas (*n* = 18) included patients of an older age; in this group we managed to achieve total shutdown in only 66.7% cases (*n* = 12), mainly in the carotid body type subtype. In this group, in two cases coils were used together with NAEM to occlude a large feeding vessel and reduce the blood flow. In 11.1% of cases, due to the inability to conduct a balloon catheter along convoluted afferents, a Headway microcatheter was used for distal catheterization. In the same group, there was a single (4.3%) complication detected by us on magnetic resonance imaging (MRI) in the form of a small area of cerebral ischemia, which was clinically asymptomatic.

In addition, we attempted to draw a parallel between the physicochemical properties of embolic agents and their operational characteristics. The operational features of the embolic agents employed in the displayed series are outlined in Table 2.

The gel-like consistency enabled ONYX, SQUID, and PHIL to attain good control, high penetration, and total distribution.

### 2.2. Illustrative Cases

As a demonstration, we present three clinical cases that most clearly demonstrate the advantages of NAGLEMs compared to other embolic agents.

#### 2.2.1. Case #1

A 35-year-old male patient was admitted to the clinic with complaints of a cosmetic defect in the lower lip area. He has been ill since birth; 18 years ago he was diagnosed with AVM of the lower lip and microsurgical removal was performed. Over the past three years, the growth of formation has been noted. Angiography revealed a relapse of arteriovenous malformation with afferents from the left facial artery (Figure 3). In three years, he underwent several stages of treatment: endovascular embolization with gelatin sponge and embolization with pure ethanol, without effect. The patient underwent endovascular embolization of the AVM of the lower lip with 5 mL of non-adhesive composition ONYX18. Arteriovenous malformation was shut down totally (Figure 4, Figure 5 and Figure 6). The patient was directed to the next stage of surgical treatment—AVM removal by cosmetic surgeons (Figure 7). The mRs score at discharge was 1.

#### 2.2.2. Case #2

A 48-year-old male patient was admitted to the clinic with complaints of a subcutaneous volume formation in the lower jaw area on the left. Earlier, in another clinic, an attempt was made to embolize the AVM using microcoils. Angiography was performed; filling of the AVM of the soft tissues of the face in the area of the angle of the lower jaw on the left was noted. The presence of separable coils in the afferent from the previous operation was noted; however, the AVM was filled through the coils (Figure 8 and Figure 9). Total embolization of the AVM of the face soft tissues with 7.5 mL of non-adhesive composition ONYX18 was performed. Arteriovenous malformation was totally shut down (Figure 10). There were no complications. The mRs score at discharge was 0.

#### 2.2.3. Case #3

A 58-year-old male patient was admitted to the clinic complaining of a palpable formation in the left submandibular region. When angiography was performed, the filling of the vascular network of the tumor of the left submandibular region from the branches of the left occipital artery, as well as the muscular branches of the left vertebral artery, was noted (Figure 11). The patient underwent intravascular embolization of the paraganglioma with 7.5 mL of non-adhesive composition ONYX18 (Figure 12 and Figure 13). The vascular network supplying the paraganglioma was partially switched off (Figure 14). There were no complications. The mRs score at discharge was 1.

## 3. Discussion

At present, the following materials are used in the endovascular treatment of hypervascularized formations of the head and neck: microcoils, polyvinyl alcohol-based particles, cyanoacrylate-based adhesives, and NAGLEMs.

Intravascular embolization of hypervascularized formations of the head and neck in the vast majority of cases is used as a preparatory stage before microsurgical removal, thereby reducing the risks of intraoperative hemorrhagic complications, as well as increasing the radicality of the formations’ removal [6]. The method of preoperative embolization consists of filling the vascular network of the formation with an embolic agent in order to stop the blood flow through the main afferents of the formation to facilitate its surgical removal. Due to the presence of a highly developed network of vessels in such formations, the main aim of embolization is to fill the entire volume of the vascular network as much as possible. Most vividly in our series, this was demonstrated by the Face AVMs group and case #1. It was the adequate total filling of the entire AVM that allowed the complete removal of the hypervascularized formation; only NAGLEMs were able to cope with this task and previous attempts were unsuccessful. An unfilled part of the vascular network leads to the development of collateral blood flow, thereby increasing the risks of surgical complications, and may become a predictor of relapse [16,26]. (An unfilled part of the vascular network of the formation, with insufficient penetration of the embolic agent into the small vessels of the formation, increases the risks of intraoperative complications, and also leads to the development of collateral blood flow, thereby being a predictor of relapse.) Due to the insufficient amount of scientific justification, embolization has not yet been considered as the final method of treatment; however, there are investigations showing the effectiveness of such interventions in high-risk patients [27] Case #2 demonstrated the possibility of total shutdown of the AVM without subsequent removal, i.e., embolization in this case was an independent and adequate method of treatment. Previous attempts at embolization with coils also did not lead to filling of the entire vascular network, which NAGLEMs subsequently coped with.

The process of successful embolization of hypervascularized formations depends on many factors, such as the experience of the surgeon, the somatic status of the patient, a competent anesthesia manual, equipment of the operating room, and, above all, the properties of embolizing agents. Currently, there are no randomized studies comparing various embolizing agents in the treatment of hypervascularized formations of the head and neck; however, reviews and articles have appeared that allow the effect of the properties of agents on the clinical outcome to be assessed [26].

The method of embolization of hypervascularized formations using microcoils solely is not used, since they can occlude only the proximal part of the afferents while maintaining the filling of the main internal network of the hypervascularized formation by the collateral blood flow. There are few reports on the use of microcoils in preoperative embolization of paragangliomas and AVMs of soft tissues of the head. The main described advantage of preoperative embolization with microcoils is the ability of palpatory detection of them during surgery and the absence of risks of distal embolism [7]. The disadvantages include the high cost of microcoils [28]. In our opinion, microcoils alone (without NAGLEMs) cannot be used for embolization due to the preservation of blood flow through smaller afferents and microcirculatory network, which was well demonstrated by the AVM of the face group and in case #2.

The method of embolization via particles based on polyvinyl alcohol is based on the gradual introduction of a prepared solution of particles of different sizes, from 100 to 1000 microns, into the vascular network of the neoplasm, from smaller to larger [29]. When using this technique, it is almost never possible to embolize the vascular network completely. Even with angiographically complete shutdown of blood flow in the formation, it is necessary to remember the smaller vessels where particles cannot get due to their size [30]. That is why collaterals begin to form, through which the blood supply of the formation continues [31]. According to Pauw BK et al., the recanalization of paragangliomas of the jugular foramen area reaches 30% as soon as 9 days after embolization with PVA particles [32].

During embolization, it is necessary to take into account the presence of potentially dangerous anastomoses between ICA and ECA, as well as ECA and VA, to prevent undesirable phenomena due to accidental migration of particles. Since PVA particles are X-ray negative, delivery is carried out with an iodine-containing contrast agent [33]. Therefore, after contrast elimination, it is impossible to estimate the pools of probable random migrations [30].

The lack of control of distal embolism, the high frequency of recanalization after embolization, and the reduced penetration into the tissue (compared with liquid embolizing agents) suggest that the spectrum of application of PVA particles in surgery of hypervascularized formations should be significantly narrowed.

Cyanoacrylate-based adhesives were the first among the liquid embolic agents to appear [23]. Rapid polymerization of cyanacrylates (from a few seconds) is the main problem of this group of embolizing agents when filling the vascular network of hypervascularized formations. Often the embolic agent does not have time to fill the vascular network of the formation completely before its polymerization [14]. The amount of lipiodol mixed with cyanoacrylate directly affects the polymerization rate; therefore, the degree of dilution is determined by the blood flow rate and the depth to which the penetration of glue is desirable [34]. As a final result, embolization of hypervascularized formations by cyanoacrylates leads to their continued growth due to collaterals located distal to the afferents turned off by glue, and, thus, a dissonance is formed between the absence of afferent vessels available for further embolization and the formed extensive vascular network of hypervascularized formation. This was demonstrated in both our groups (both with facial AVM and with paragangliomas) where previous embolizations with cyanoacrylates did not lead to any result. In such a situation, there are no embolization possibilities and the treatment of such formations passes to microsurgeons, while the risks of intraoperative bleeding remain. For many years before the advent of non-adhesive compositions, cyanoacrylates remained the only available liquid embolic agents in the treatment of hypervascularized formations of the head and neck.

NAGLEMs (ONYX, SQUID, PHIL) are widely used for intravascular embolization of hypervascularized formations. All of the above NAGLEMs have certain potential advantages and disadvantages. Despite their structural differences, ONYX, SQUID, and PHIL have similar properties. All three non-adhesive compositions clog blood vessels as a result of “precipitation”; this mechanism is often compared to the solidification of a lava flow [14]. DMSO is used as a solvent in all three agents. However, it would appear that gelation occurs in biological tissues in addition to the mechanisms of absorption, which allows us to refer to these compositions as gel-like agents. While cyanoacrylates polymerize over a period of several seconds to several minutes, in non-adhesive compositions, the polymerization mechanism can take up to 30–40 min, depending on the size of the embolized blood vessels and the speed of blood flow in them. This aspect provides more controlled embolization, longer injection time and, consequently, better penetration and filling of the target formation, but also contributes to undesirable diffusion into normal arteries. A significant difference between SQUID and ONYX is the smaller size of tantalum powder granules [15,17]. The smaller size of tantalum granules is aimed at increasing the uniformity of radiopacity and improving visibility with longer injections [35].

Compared to ONYX and SQUID, PHIL flows forward more like a column rather than like the above behavior with precipitation from outside to inside. PHIL has a fairly high embolic capacity. Compared to SQUID 18 and ONYX18, smaller volumes of PHIL are required for the same degree of embolization [23]. All three agents have several versions of different viscosities in their lines (ONYX18, ONYX20, and ONYX34; SQUID12 and SQUID18; PHIL25, PHIL30, and PHIL35), which significantly expand the boundaries of their use. Depending on the speed of the shunting process and the volume of formation, it is possible to select the necessary version of the embolizing agent that will provide faster and more effective penetration into the target vascular network [23]. In our series detailing the treatment of patients with hypervascularized masses, we were able to develop the concept of curative embolization through the use of NAGLEM distal penetration into the smallest vessels, along with the distribution of non-adhesive composition throughout the tumor stroma. This approach is further exemplified in case #3, where Figure 12, Figure 13 and Figure 14 demonstrate its success. This endovascular embolization served as a standalone treatment since the vasculature was entirely disconnected, and the paraganglioma did not necessitate any additional removal.

The gel-like consistency enables the use of embolic agents with varying viscosities to achieve precise control, high penetration, and complete distribution (Table 2). However, there is limited understanding of the mechanisms of sol–gel transformation and precipitation during the polymerization of NAGLEMs in biological tissues, as well as the dependence of these processes on temperature at both local and delivery system levels. Further study of these processes will enhance the predictability and effectiveness of embolization technology.

## 4. Conclusions

The benefits of using NAGLEMs in treating hypervascularized head and neck neoplasms currently stem from their key characteristics: non-adhesiveness and delayed, predictable polymerization (takes place after 30–40 min); efficient dispersion and infiltration through the neoplasm’s vascular-distal duct; manageable process control mainly due to the range of embolization viscosities; and outstanding X-ray visualization. Currently, NAGLEMs are the sole effective embolic agents for fully and adequately occluding the vascular bed of hypervascular volumetric masses, taking into consideration all of these characteristics.

## 5. Materials and Methods

### 5.1. Study Design

This was a case–control, non-randomized study. The case series was designed to evaluate the effectiveness and results of the use of these NAGLEMs in treatment of patients in our clinic, and to compare them with other embolic agents. The study was conducted at the North-Western district scientific and clinical center named after L. G. Sokolov Federal Medical and Biological Agency & Almazov National Research Medical Center in St. Petersburg, Russia. Patients were enrolled in the study from November 2015 to May 2023.

We analyzed the technical features, effectiveness, and safety of endovascular treatment of two groups of hypervascularized pathological formations localized in the soft tissues of the head and neck in 23 patients operated on from 2015 to 2023. The age of the patients ranged from 29 to 76 years and averaged 55 ± 13 (M ± SD) years. In technical features, the type of SQUID embolizing agent was evaluated in 12 cases (52.2%), ONYX in 8 (34.8%), PHIL in 1 (4.3%), and a combination of ONYX + SQUID in 2 (8.7%). The types of delivery microcatheter used were DMSO-compatible balloon catheters Scepter C, XC (Microvention), used for 91.3% (*n* = 21), and Headway microcatheters (Microvention), used for 8.7% (*n* = 2). The effectiveness was assessed based on the degree of shutdown of the vascular network of formations (total shutdown was achieved in 73.9% (*n* = 17) of cases, and partial shutdown in 26.1% (*n* = 6)), and the outcome on the mRs scale, the average of which was 1. Safety was assessed by the presence of complications, which developed in 1 case (4.3%), and changes in postoperative MRI, which was performed in all patients at the time of discharge from the clinic.

### 5.2. NAGLEM Embolization Technique

Embolization was carried out according to the following method. After performing angiography in standard projections, afferents to the formation were determined and blood flow was assessed along the anterior and posterior communicant arteries. A DMSO-compatible microcatheter or balloon catheter was inserted into the identified afferents and microangiography was performed, with an assessment of the so-called dangerous anastamoses. Then, through the same catheters and balloons for the delivery of NAGLEMs, embolization of hypervascularized formations was carried out with NAGLEM injection to achieve maximum distal penetration. In cases where only DMSO-compatible microcatheters were used, surgical intervention was lengthened due to the time required for formation due to reflux and the formation of a proximal “plug”. In one case coils were used according to the “Pressure Cooker Technique” described earlier for AVM of the brain [36]. In some cases, a DMSO-compatible balloon catheter was installed to protect against cerebral vascular embolism in the internal carotid artery. After the introduction of NAGLEMs, a control angiography was performed in which the degree of embolization was assessed.

### 5.3. Statistical Analysis

Statistical analysis was carried out using the StatTech v. 3.1.10 program (developed by StatTech LLC, Moscow, Russia).

Quantitative indicators were evaluated for compliance with the normal distribution using the Shapiro–Wilks criterion (with the number of subjects less than 50) or the Kolmogorov–Smirnov criterion (with the number of subjects more than 50).

Quantitative indicators having a normal distribution were described using arithmetic averages (M) and standard deviations (SD), and the boundaries of the 95% confidence interval (95% CI).

In the absence of a normal distribution, quantitative data were described using the median (Me) and the lower and upper quartiles (Q_1_–Q_3_).

Categorical data were described with absolute values and percentages. Comparison of the two groups by a quantitative indicator having a normal distribution, provided that the variances were equal, was performed using the Student’s *t*-test.

Comparison of three or more groups by a quantitative indicator having a normal distribution was performed using one-factor analysis of variance; a posteriori comparisons were carried out using the Tukey criterion (provided that the variances were equal).

The comparison of the two groups by a quantitative indicator, the distribution of which differed from the normal one, was performed using the Mann–Whitney U-test.

Comparison of three or more groups by a quantitative indicator, the distribution of which differed from the normal one, was performed using the Kruskal–Wallis criterion; a posteriori comparisons were performed using the Dunn criterion with the Hill correction.

The comparison of percentages in the analysis of four-field conjugacy tables was performed using the exact Fisher criterion (with values of the expected phenomenon less than 10).

The comparison of percentages in the analysis of multipole conjugacy tables was performed using Pearson’s chi-square criterion. The direction and closeness of the correlation between the two quantitative indicators were assessed using Spearman’s rank correlation coefficient (with a distribution of indicators other than normal). A predictive model characterizing the dependence of a quantitative variable on factors was developed using the linear regression method.

The construction of a predictive model of the probability of a certain outcome was carried out using the logistic regression method. The Nigelkirk coefficient R^2^ was used as the measure of certainty that indicated the part of the variance that could be explained by logistic regression.

A full statistical analysis can be seen in the Appendix A.

To assess the diagnostic significance of quantitative signs in predicting a certain outcome, the method of analysis of ROC curves was used. The separating value of the quantitative attribute at the cut-off point was determined by the highest value of the Yuden index. An analysis of the available literature was also performed. The PubMed library was searched for articles using the following keywords: “AVM of the head and neck, embolization of extracranial AVMs, embolization of paragangliomas, embolic agent”. Criteria for articles’ inclusion: a series of patients with hypervascularized formations of the head and neck and the use of embolizing agents in their treatment. The exclusion criteria included percutaneous embolization. A total of 7 articles were selected [3,4,5,9,29,37,38].

## Figures and Tables

**Figure 1 gels-09-00954-f001:**
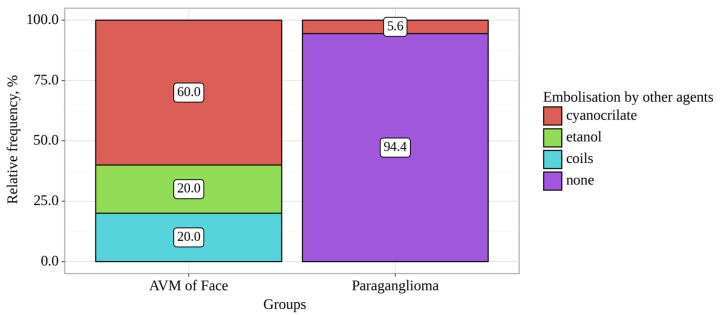
Analysis of embolization by other agents conditioning on groups.

**Figure 2 gels-09-00954-f002:**
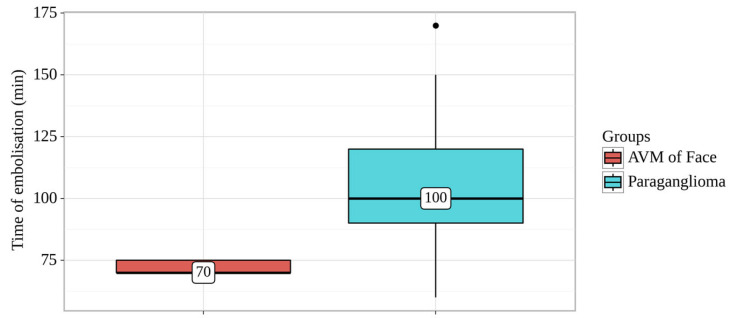
Analysis of time of embolization conditioning on groups.

**Figure 3 gels-09-00954-f003:**
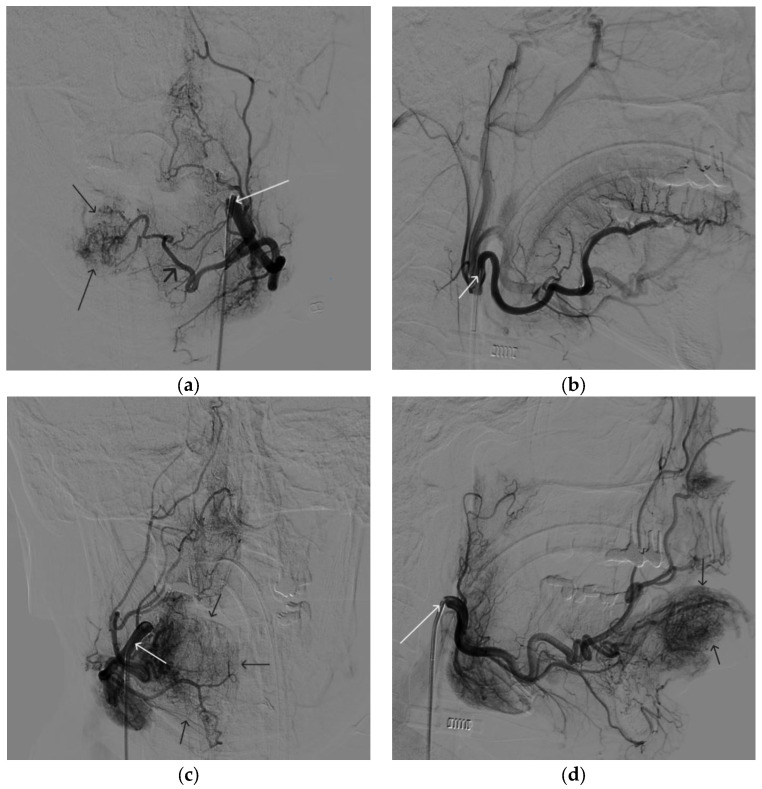
Digital subtraction angiography from the facial arteries: (**a**,**b**) straight and lateral projections on the left; (**c**,**d**) straight and lateral projections on the right (white arrows indicate catheters at the ostium of the facial arteries, black long arrows indicate filling of the lower lip AVM, black short arrow indicates the afferent AVM from the left facial artery). There is filling of the AVM from the afferent from the left facial artery.

**Figure 4 gels-09-00954-f004:**
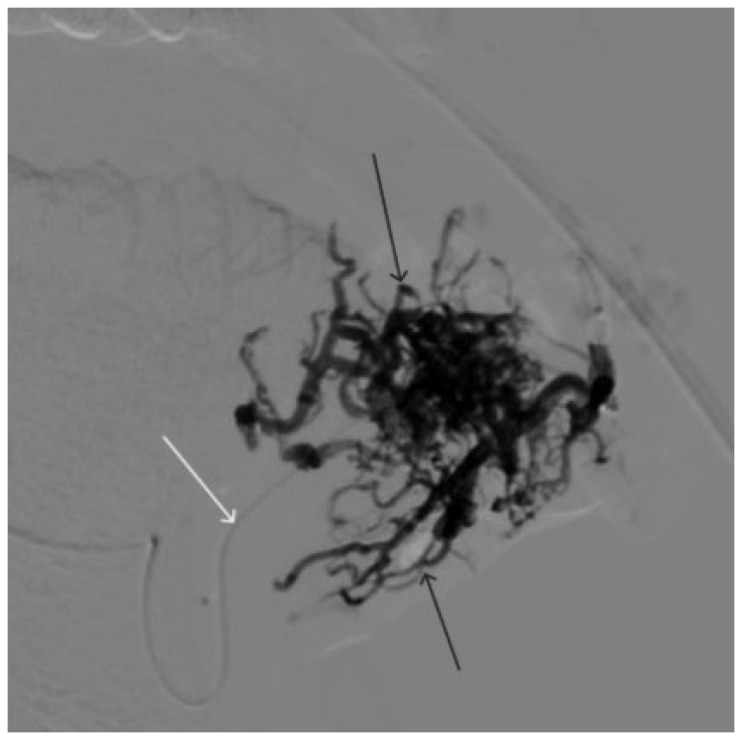
The process of introducing ONYX18 into the AVM of the lower lip (the white arrow indicates the microcatheter, the black arrow indicates the spread of embolizate along the vascular network of the formation).

**Figure 5 gels-09-00954-f005:**
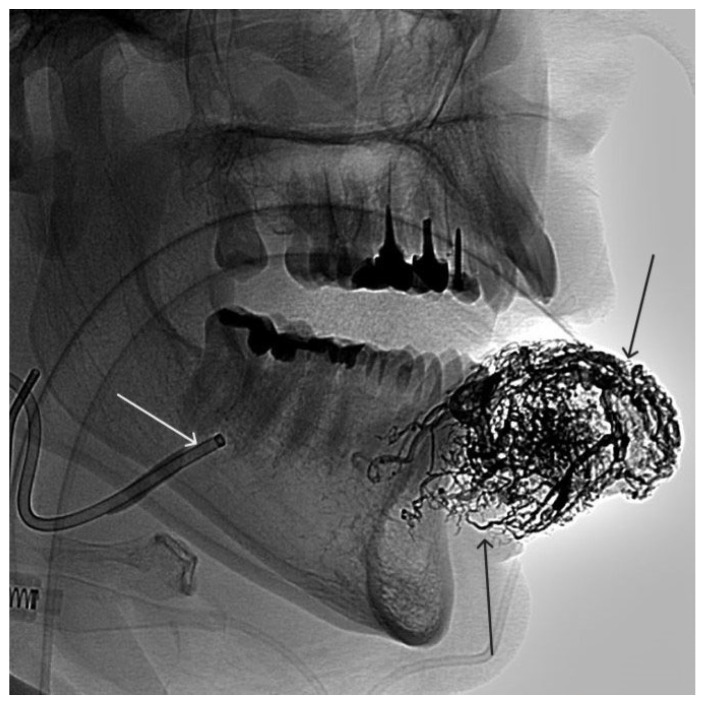
X-ray in single shot mode. The NAGLEMs cast is visualized, filling the vascular network (the white arrow indicates the guiding catheter at the ostium of the left facial artery, the black arrow indicates the cast).

**Figure 6 gels-09-00954-f006:**
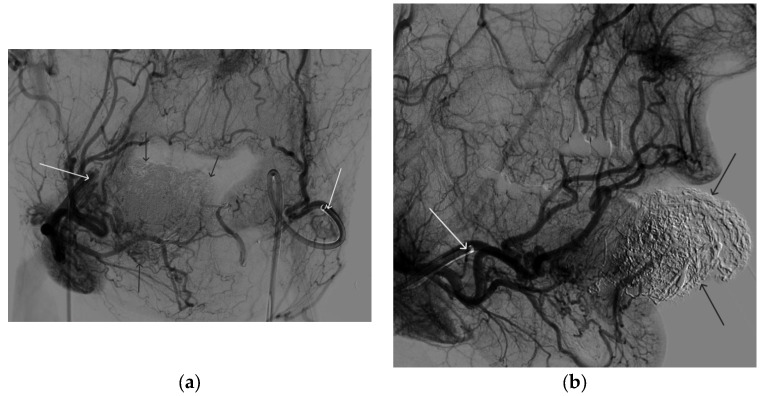
Digital subtraction angiography from the right and left facial arteries: (**a**) direct projection; (**b**) lateral projection (white arrows indicate catheters at the ostium of the facial arteries, black arrows indicate embolizate cast). The absence of AVM contrast is noted.

**Figure 7 gels-09-00954-f007:**
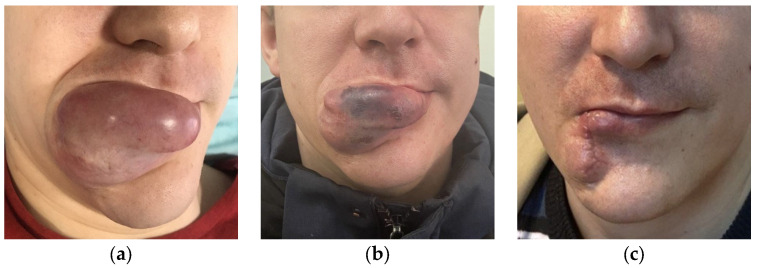
(**a**) The appearance of the AVM before embolization, (**b**) the appearance of the AVM after embolization, (**c**) the patient after surgical removal of the AVM.

**Figure 8 gels-09-00954-f008:**
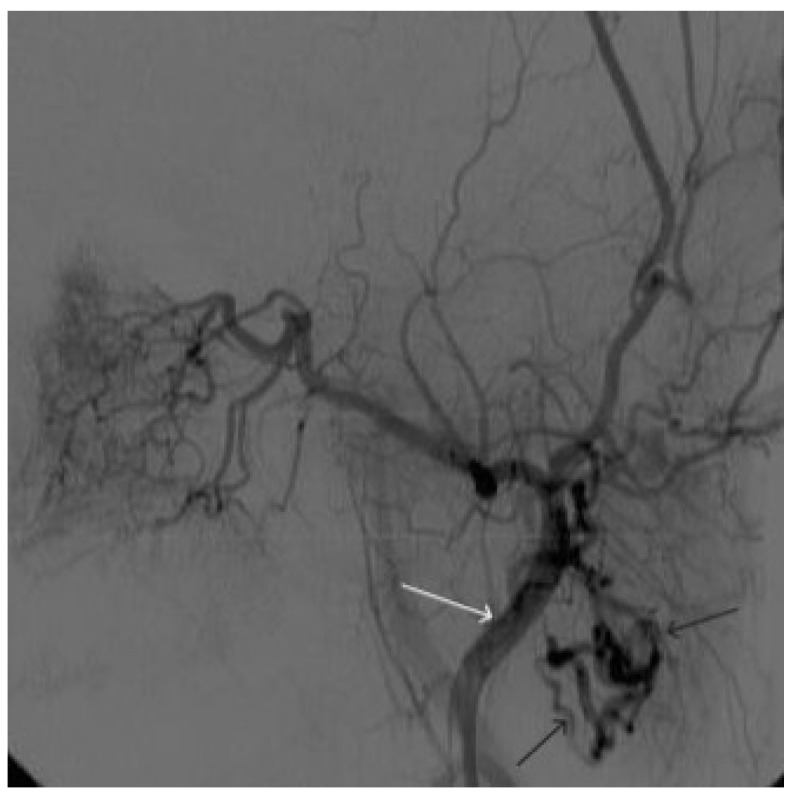
Digital subtraction angiography from the left external carotid artery, direct projection (white arrow indicates catheter in the left external carotid artery, black arrows indicate contrast AVM).

**Figure 9 gels-09-00954-f009:**
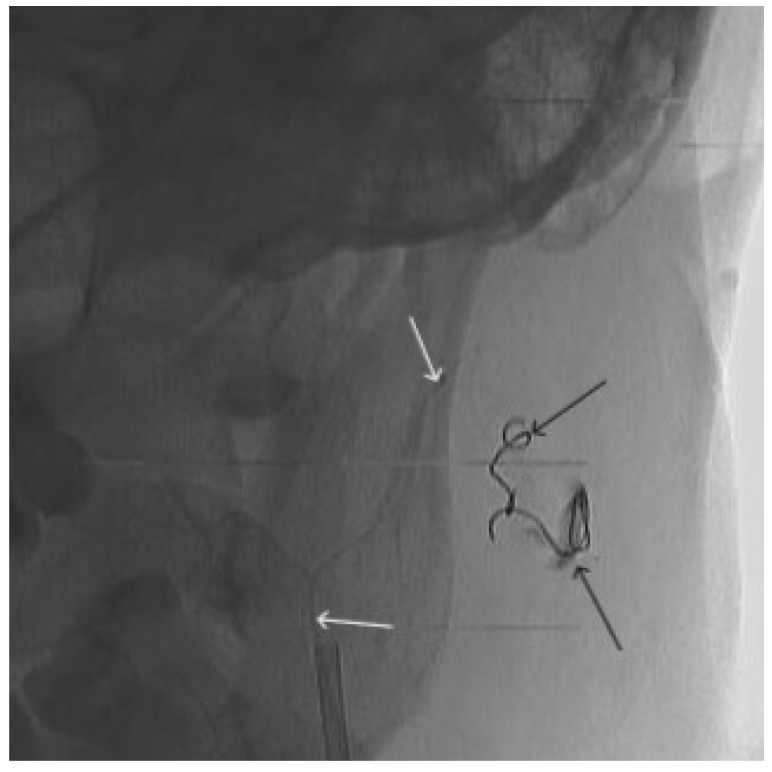
X-ray, direct projection (the white arrow indicates a microcatheter in the left external carotid artery, the black arrows indicate a complex of microcoils from the previous stage of embolization).

**Figure 10 gels-09-00954-f010:**
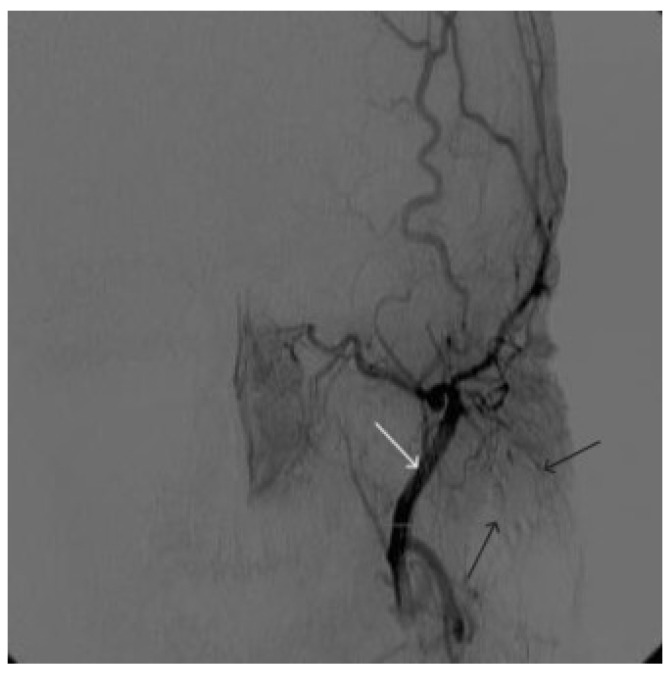
Digital subtraction angiography from the left external carotid artery, direct projection (white arrow indicates the left external carotid artery, black arrows indicate the boundaries of the filled AVM).

**Figure 11 gels-09-00954-f011:**
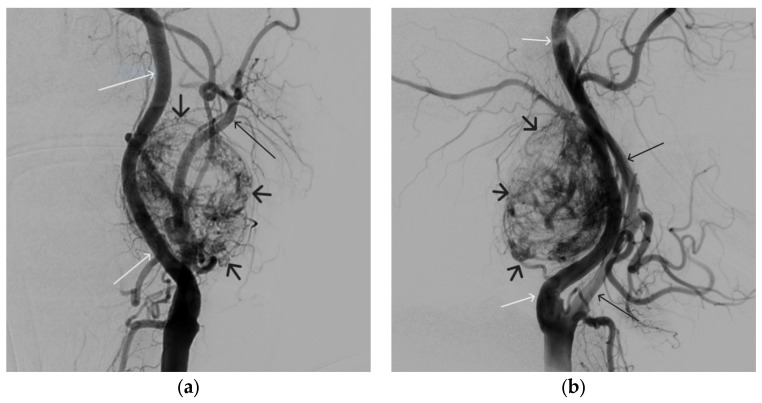
Digital subtraction angiography from the left common carotid artery: (**a**) direct projection, (**b**) lateral projection (white arrows indicate the left internal carotid artery, long black arrows indicate the external carotid artery, short black arrows indicate the boundaries of the paraganglioma).

**Figure 12 gels-09-00954-f012:**
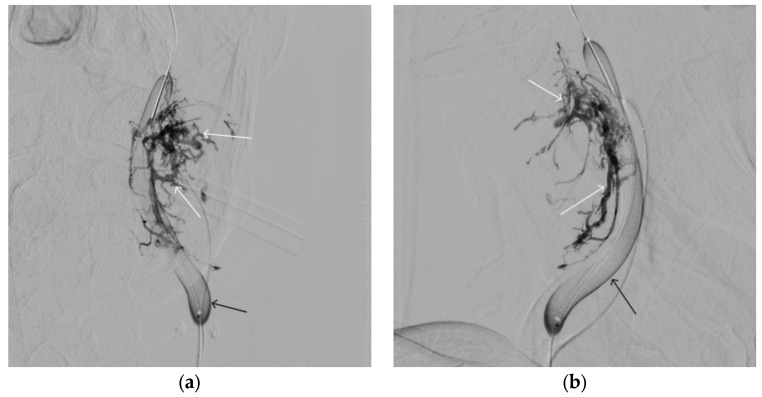
Distribution of ONYX18 along the vascular network of the paraganglioma: (**a**) direct projection, (**b**) lateral projection (white arrows indicate filling of the vascular network of the paraganglioma, black arrows indicate an inflated balloon in the left internal carotid artery).

**Figure 13 gels-09-00954-f013:**
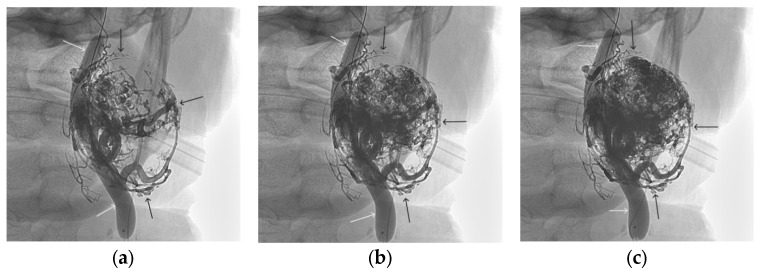
X-ray in single shot mode. (**a**–**c**) Demonstration of gradual spreading of ONYX18 (NAGLEMs cast) through the vascular network of the paraganglioma direct projections (black arrows indicate filling of the vascular network of the paraganglioma, white arrows indicate an inflated balloon in the left internal carotid artery).

**Figure 14 gels-09-00954-f014:**
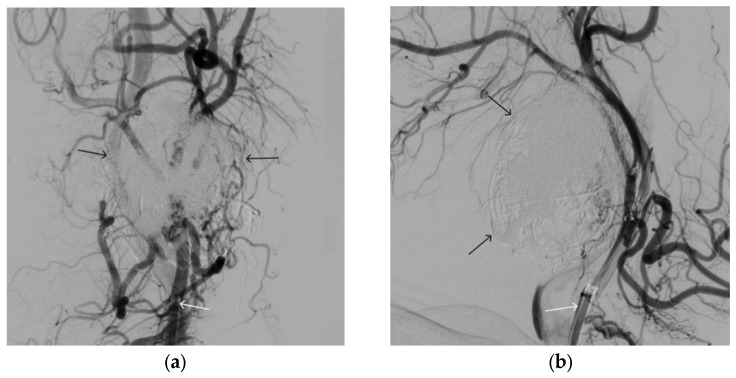
Digital subtraction angiography from the left common carotid artery: (**a**) direct projection, (**b**) lateral projection (white arrows indicate the guiding catheter in the left external carotid artery, black arrows indicate the boundaries of the embolized paraganglioma).

**Table 1 gels-09-00954-t001:** Summary of the demographic and treatment data of two patient groups treated by NAGLEM embolization.

Variables	Face AVMs (*n* = 5)	Paragangliomas (*n* = 18)
Age (years), M ± SD	42 ± 6	58 ± 12
Sex (female/male)	3/2	13/5
Number of treatment stages (Me, Q_1_–Q_3_)	2, 2–2	1, 1–2
Number of NAEM embolization steps (Me, Q_1_–Q_3_)	1, 1–1	1, 1–2
Open Surgical Interventions after embolization (%)	3 (60.0%)	1 (5.6%)
Coils while NAEM embolization (%)	none	2 (11.1%)
Total embolization (%)	5 (100%)	12 (66.7%)
Type of catheter (%)	Scepter C, XC (100%) (Microvention)	Scepter C, XC (88.9%) Headway (11.1%) (Microvention)
mRS before embolization Me, Q_1_–Q_3_	0, 0–0	1, 0–1
mRS at discharge Me, Q_1_–Q_3_	0, 0–1	1, 0–1

**Table 2 gels-09-00954-t002:** Summary of operational characteristics of embolic agents based on their physicochemical properties (from the literature and own cases). (“-”–none; “+“–poor; “++“–moderate; “+++“–excellent).

Characteristics	Embolic Agents				
	Cyanoacrylates (Glue)	Sclerosing Agents (Ethanol)	Particles	Microcoils	NAGLEMs
**Radiopacity**	++	-	+	+++	+++
**Penetration**	++	+++	++	-	+++
**Distribution**	++	-	+	-	+++
**Controllability**	+	-	-	+++	++

## Data Availability

The data presented in this study are available on request from the corresponding author. The data are not publicly available due to the privacy of the patients who assisted in the research.

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
