# Peer review of "The Advantages of Non-Adhesive Gel-like Embolic Materials in the Endovascular Treatment of Benign Hypervascularized Lesions of the Head and Neck"

_gels, 2023, doi:10.3390/gels9120954_

Round 1

Reviewer 1 Report

Comments and Suggestions for Authors

This paper reported several cases using the gel for the endovascular treatment of lesions in human, the cases are interesting, and the references are relevant. Only one comment:

Could the authors make a table to compare the advantages and disadvantages of gels, surgical excision, ? 

Author Response

Thank you for the evaluation of our work.
According to your recommendations, we have added a table 2 comparing different embolisates in terms of basic explantation characteristics. More details on advantages and disadvantages are given in the discussion. It seems to us that a comparison of different embolic agents would be more informative and more illustrative. It is difficult to directly compare gel embolisation with surgical removal in small and heterogeneous cases. Furthermore, gel embolisation is performed both immediately before and instead of surgical removal. This may become a separate task in a larger sample. We are ready for further work.

Thank you again for your valuable comments.

Reviewer 2 Report

Comments and Suggestions for Authors

1. The introduction is too poor.

2. What is non adhesive gel like embolic materials. The authors should define them.

3. The introduction has to be more informative. It should discuss the problem, its aspects, etc. Then the traditional techniques that are used and their limitations should be mentioned before mentioning the proposed study aim.

4. The statistical analysis should be clarified on the charts. 

Comments on the Quality of English Language

Minor editing

Author Response

Thank you for the evaluation of our work,
According to your recommendations:

  1. We have rewritten the introduction and added links to 10 other articles that we mention in the discussion.
  2. We introduced the definition of NAGLEMs in the Introduction section, and it now looks like this:

Non-adhesive gel-like embolic materials (NAGLEMs) are a group of biocompatible co-polymer compositions that dissolve in dimethyl sulfoxide (DMSO). When in contact with blood, they solidify into a sponge-like gel, forming a rigid, plastic-like cast.  Considering the lack of a clear understanding of the sol-gel transition conditions in biological tissues, these embolic materials can rightly be classified as gel-like, apparent-ly forming at various stages structures similar to thermoreversible xerogel.

  1. In the introductory section, we focused on explaining the definition of NAGLEMs and its rationale, but did not go into detail about the characteristics of other embolic agents, as these are given in other review articles in the reference list. In addition, we have added Table 2, which compares the different types of embolic devices in terms of explantation characteristics. If the reviewer feels it is necessary to add a description of other embolic agents, this can be done, but it would lengthen the introductory section.
  2. Statistical analyses are included in the supplementary material, and Figures 1 and 2 present the most illustrative charts. 

We are ready for further work.

Thank you again for your valuable comments.

Reviewer 3 Report

Comments and Suggestions for Authors

The authors report on the application of NAGLEMs to hypervascularized head and neck neoplasms when compared to other embolizing substances. This study is interesting because it shows the superiority of the gel form for specific conditions and provides evidence of the usefulness of gels in the medical field. Whereas the reviewer thinks the authors’ study in this manuscript is quite interesting, suggestive, and well-organized, some descriptions are insufficient. The authors’ manuscript is not suitable for publication in “Gels” in the present form. From these considerations, the reviewer recommends accepting for publication in “Gels,” if the following issues are resolved.

1)      The introduction needs to be more comprehensive to understand the authors' study; a description of NAGLEMs and NAGLEMs as gels should be included, along with the historical background. It is too short.

2)      The " gels" of the NAGLEMs used by the authors are important for the purpose of their study when compared to other forms of embolizing substances (coils, glue, and particles). What kind of appearance and elastic modulus do these NAGLEMs have, and what category of gels are they? What is the difference between ONYX, SQUID, and PHIL as gels?

3)      What data should the reader understand about whether NAGLEMs (ONYX, SQUID, PHIL) are gels? What molecules are NAGLEMs made of?

4)      Not only should the authors show the predominance of the gel state of the NAGLEMs used, but also a discussion of the correlation between the molecular structure of the NAGLEMs and the results.

5)      A discussion of the results obtained for ONYX, SQUID, and PHIL should be added, considering their material characteristics and results. If it is only a case study, it should be reported in an appropriate medical journal. If it is reported in "Gels," further consideration should be given to why the gel condition was good.

Author Response

Thank you for the evaluation of our work,
According to your recommendations:

1. We have rewritten the introduction and added links to 10 other articles that we mention in the discussion. Non-adhesive gel-like embolic materials (NAGLEMs) are a group of biocompatible co-polymer compositions that dissolve in dimethyl sulfoxide (DMSO). When in contact with blood, they solidify into a sponge-like gel, forming a rigid, plastic-like cast.  Considering the lack of a clear understanding of the sol-gel transition conditions in biological tissues, these embolic materials can rightly be classified as gel-like, apparent-ly forming at various stages structures similar to thermoreversible xerogel.

  1. In the introductory section, we focused on explaining the definition of NAGLEMs and its rationale, but did not go into detail about the characteristics of other embolic agents, as these are given in other review articles in the reference list. In addition, we have added Table 2, which compares the different types of embolic devices in terms of explantation characteristics. We discuss the physicochemical properties in the Introduction and Discussion sections, where we give the composition and viscosity parameters and refer to the papers where they are discussed. We have not presented these properties separately in the form of tables, as they are given in the previous papers we have cited. However, if the reviewer strongly recommends this, we agree. In this paper, we have tried to highlight and draw parallels between physical and chemical properties and exploitation characteristics, which are presented in Table 2.
  2. Compositional data and chemical formulae are given in the Introduction section; if the reviewer feels that a graphical representation of the formulae is necessary, we will provide it, but these formulae were given in the papers to which we refer.
  3. We have attempted to answer this question with Table 2 and in the Discusion section. If the reviewer feels that additional comparison tables between physicochemical properties and explanatory properties are needed, we will add them. however, the physicochemical properties and formulae are described in other papers.

  1. In this article we do not compare the efficacy of ONYX, SQUID and PHIL with each other, the main aim is to show the efficacy of the embolysate group - NAGLEMs compared to other embolic agents for pathological hypervascularised masses.

We are ready for further work.

Thank you again for your valuable comments.

Round 2

Reviewer 2 Report

Comments and Suggestions for Authors

The comments have been addressed successfully by the authors.

Comments on the Quality of English Language

Minor editing is required

Author Response

Dear reviewer, 
Thank you for your appreciation of our manuscript and your recommendations.
We have made the linguistic corrections to the article. And we hope to publish it soon.
Thank you again, with best wishes
The author team.

Reviewer 3 Report

Comments and Suggestions for Authors

As shown in the revised manuscript, some issues suggested by the reviewer were resolved.

The reviewer believes that the authors' findings will contribute to the advancement of materials science and chemistry of gels.

The reviewer recommends accepting the revised manuscript for publication in “Gels."

3) "Compositional data and chemical formulae are given in the Introduction section."; The authors should move these data to the experimental part.

Author Response

Dear reviewer, 
Thank you for your appreciation of our manuscript and your recommendations.
We have made the text transfer from the introduction section as per your recommendations.
Thank you again and best wishes
The author team.